# Children’s Attitudes and Behaviors about Oral Health and Dental Practices

**DOI:** 10.3390/healthcare9040416

**Published:** 2021-04-04

**Authors:** Sara Costa Fernandes, Ana Louceiro, Luísa Bandeira Lopes, Francisco Esteves, Patrícia Arriaga

**Affiliations:** 1Centro de Intervenção e Investigação Social (CIS-Iscte), ISCTE-Instituto Universitário de Lisboa, 1649-026 Lisboa, Portugal; patricia.arriaga@iscte-iul.pt; 2Faculdade de Ciências Médicas, Universidade Nova de Lisboa, 1169-056 Lisboa, Portugal; ana.f.louceiro@edu.nms.unl.pt; 3Clinical Research Unit (CRU), Centro de Investigação Interdisciplinar Egas Moniz (CiiEM), Dental Pediatrics Department, Egas Moniz—Cooperativa de Ensino Superior, 2829-511 Almada, Portugal; luisabpmlopes@gmail.com; 4Department of Psychology and Social Work, MidSweden University, 831 25 Östersund, Sweden; francisco.esteves@miun.se

**Keywords:** oral health, dental practices, oral health promotion

## Abstract

The present study sought to contribute to a better understanding of children’s attitudes and behaviors regarding oral health and dental practices. The sample was composed of 101 children (8–10 years), collected from several schools in the Lisbon metropolitan area. Our main goals were to collect a survey of information about the beliefs, attitudes, habits and knowledge of children about oral and dental issues in order to not only have an overview of them but also to serve as a basis and a starting point for the development of intervention programs to increase positive attitudes and behaviors related to oral health and promote greater knowledge about these subjects. In general, children reported positive behaviors regarding dental habits and oral hygiene practices. Children’s opinions and beliefs about dentists were also globally positive; however, the results suggested that younger children reported more positive attitudes, emotions and previous experiences. Regarding children’s knowledge about these dental issues, the results were quite negative and worrying, ruled by ignorance and incorrect beliefs and behaviors. In sum, all the results and conclusions of this study may contribute to the development of educational programs within the scope of the promotion of oral health and hygiene practices—“An Adventure about Oral Health.”

## 1. Introduction

Dental issues related to oral health and hygiene practices are crucial challenges for all modern societies, especially for children. Fear and anxiety toward dental issues and going to the dentist affects a considerable proportion of children and adolescents [1], with a huge negative impact on their future behaviors. According to the literature, the prevalence of dental fear and anxiety is recognized as a public health dilemma [2], potentially caused by both individual and external origins, in particular subjective (e.g., quantity and quality of previous dental visits, lack of information), social (e.g., influence of parents or peers experiences or attitudes) and contextual factors (e.g., lack of clinical skills, dental settings and sounds, medical procedures) [1]. The negative impact of fear and anxiety on children’s future dental behaviors has also been recognized, in particular on uncooperativeness or even prolonged avoidance of dental treatment [2]. For example, patients with dental fears tend to miss three times more appointments than other patients [3]. On the other hand, avoiding dental treatment due to fear and anxiety exacerbates problems related to patients’ oral health. From the perspective of health professionals, it would be even more difficult and time-consuming treating anxious patients, and it has been vital for dentists to construct an environment that alleviates patient worries, fears, and anxiety [2].

One way to reduce children’s anxiety is the use of sedation. However, sedating children before dental procedure carries high risks for both anesthesiologists and patients [4], although, in cases of higher levels of dental fear and anxiety, conscious sedation or even general anesthesia tend to be recommended and administered [5].

Children’s attitudes and behaviors toward healthcare are globally negative and dis-criminatory and are guided by the lack of education and biased beliefs [6]. Providing children with information about health promotion is expected to afford greater recognition, understanding and more healthy and positive behaviors.

The present project seeks to collect information to better understand children’s knowledge, attitudes and behaviors regarding oral health, as well as their behaviors and habits related to these issues. Although our study is exploratory, we expected that children would report some lack of knowledge and inaccurate information about dental care. This outreaching goal of the study is to understand how we can develop educational interventions in oral and dental health promotion programs by addressing children’s misconceptions about dental care.

## 2. Materials and Methods

### 2.1. Settings and Ethical Aspects

The study was conducted at two different schools (a public and a private school) located in the Lisbon metropolitan area from May to June 2019. The project was approved by the Ethical and Board Committees of both university (Ethical Committee approval 45/2019) and elementary schools, as well as by the Ministry of Education (School Survey No. 0474200006). Informed consent for parents was requested and children also provided assent, and all agreed to participate. All participants were guaranteed anonymity. Children were excluded if they were non-Portuguese speakers or had a developmental delay. These exclusion criteria were important to guarantee that all participants had the necessary skills to understand the questions and report their responses in the formats that were provided. However, no child was excluded from the study due to these exclusion criteria. Data were collected in both public (73.3%) and private schools (26.7%) in the Setubal district.

### 2.2. Participants

The sample (*N* = 101; male = 49; female = 52) was collected in several classroom settings, and it is composed of children between 8 and 10 years (*X* = 9.11 years; *SD* = 0.79) in the regular education levels (third and fourth grades). From the total sample, 55.4% were in the third grade and the remaining 44.6% in the fourth grade. Each child was asked to complete a self-report survey individually.

The age range between 8 and 10 years was chosen to ensure that all the samples would be at the same concrete operational stage of development, according to Piaget’s theory (1963), and the children will have similar interests and experiences.

### 2.3. Measures and Procedures

Demographic data. Children’s gender, age and level of education, as well as some knowledge, behaviors and perceptions related to dental issues, were obtained through a survey.

Dental and oral hygiene habits. Participants were asked about their dental hygiene habits: “How many times per day do you brush your teeth?” (response scale ranged from 0 to more than 3 times per day), “Do you usually brush your teeth every day?” (yes/no), “Do you brush your teeth with toothpaste?”, “Do you floss?”, “After brushing your teeth, do you use mouthwash?”, “Do you have an adult helping you brush your teeth?” (response options were 1 = yes, 2 = sometimes and 3 = no).

Previous dental problems and experience with dental treatments. Participants were asked about previous dental and oral problems and experience with treatments: “Have you ever had a cavity?”, “Have you ever had pain in your gums?”, “Have you ever bled from your gums?”, “Have you ever had a tooth taken out?”, “Have you ever been to the dentist?”, “Have you ever had a bad experience at the dentist?” Responses to these questions were given using “Yes” or “No” options.

Opinion about dentists. Participants were asked about their opinion regarding dentists. Responses were given on a scale ranging from 1 (“I don’t like them”) to 3 (“I like them”). A scenario was also presented in which they had to imagine having a problem in their teeth and going to the dentist. They were then asked if after going to the dentist they got better, worse or remained the same.

Emotions. Children’s emotions about dental practices (e.g., going to the dentist) were assessed using the response format of the Self-Assessment Manikin (SAM) scale. SAM is composed of five graphic figures (mannequins) in each one of the following two dimensions: (1) Valence, ranging from a happy to an unhappy mannequin; (2) Arousal, ranging from a highly aroused to a calm mannequin. Participants were presented with a scenario in which they visited the dentist and asked how happy/sad and nervous/calm they would feel. They responded using two self-assessment manikin scales [7]. The first consisted of five images representing a human body and its face ranging from a “Very happy” face to a “Very sad” face. The second consisted of five images in a row, representing a human body with decreasing degrees of arousal, ranging from “Very nervous” to “Very calm.” After, participants were presented with a scenario in which they had a toothache and went to the dentist. They responded using two self-assessment manikin scales similar to those presented in the previous question. Previous studies have shown that SAM has good convergent validity and internal consistency [8,9].

Feelings toward dental and oral health. Participants were asked to think about oral and dental health and to write and draw a face representing how they felt about “going to the dentist”.

Knowledge concerning dental and oral hygiene. An additional set of questions concerning dental and oral hygiene knowledge was presented to participants. Hence, they were asked questions concerning the right amount of toothpaste one should use when brushing their teeth. The response scale was pictorial, showing four images of a toothbrush and varying amounts of toothpaste, ranging from a very small portion (considered an amount that was too little), a small portion (considered the right amount), a large portion and a very large portion (both considered excessive amounts).

Participants were also asked to write down what was the maximum number of teeth an adult can have, how many types of teeth a person could have, how many times per day one should brush their teeth and how long should that activity last.

The questionnaires were used to obtain information about children’s dental and oral hygiene habits. Previous dental problems and experiences with dental treatments, as well as opinions about dentists, were constructed using the same questions included on both questionnaires, Modified Child Dental Anxiety Scale (MCDASF) [10] and Children’s Fear Survey Schedule—Dental Subscale (CFSS-DS) [11]. Several research findings have suggested that MCDASF and CFSS-DS are both valid, reliable and have good internal consistency (0.82) to measure dental experiences in 5–12-year-old children across the world [12,13].

### 2.4. Statistical Methods

Data were analyzed by using IBM SPSS Statistics 20 for Windows.

Descriptive analyses were conducted (frequencies, percentages, means and standard deviations), as well as inferential analyses such as paired sample *t*-tests, chi-squared tests, univariate and multivariate analysis of variance (ANOVA and MANOVA, respectively) and regression analyses. The data set used in this study had very few missing values due to nonresponses; hence, they were addressed as such and were not replaced by participants’ average.

### 2.5. Sample Size

G*Power 3.2.3 software analysis [14] was used to estimate the sample size, suggesting a minimum of about 84 participants that should be recruited. More specifically, for detecting a medium effect size (f = 0.25, using Cohen’s standard effect sizes) in the analysis of variance, a sample of 84 participants was suggested. For a large size effect (f = 0.40), a sample of 210 participants should be recruited. The program also suggests for the *t*-tests a minimum of 23 participants for a medium effect (d = 0.05) and 54 participants for a large size effect (d = 0.80) [15]. Our final sample was composed of 101 children, which fit the number of participants suggested for, at least, a medium effect size.

## 3. Results

### Main Results

Dental and oral hygiene habits

Regarding participants’ dental and oral hygiene habits (Table 1), most participants brush their teeth twice a day (48.5%), every day (99.0%), without the help of an adult (89.1%). Most participants use toothpaste (98%) but do not floss (48%) nor use mouthwash (41.6%).

Previous dental problems and experiences with dental treatments

Concerning participants’ previous dental problems and experiences with dental treatments (Table 1), a large majority had been to the dentist (88.9%), 45.5% already had a cavity, gum pain (54%) and bleeding (75.2%) and a tooth taken out (68.3%). Most participants (72.4%) reported never having had a bad experience at the dentist.

Opinion about dentists

Most participants have a positive opinion about dentists (Table 2), with 72.3% saying they like them, 10.9% saying they do not like them and 16.8% answering that they do not know. Regarding the scenario in which participants had to imagine they were having a problem with their teeth and went to the dentist (Table 2), 92.1% thought they would be better afterward.

Feelings toward dental and oral health

When asked to write down what they felt about oral and dental health, 68.3% reported positive feelings (“well,” “happy,” “calm”), 18.8% reported negative feelings (“fearful,” “nervous,” “concerned”) and 12.9% reported neutral words (“normal,” “more or less”).

Emotional responses: valence and arousal

Regarding the scenario in which they had to imagine going to the dentist, most participants imagined feeling very happy (40.6%) and very calm (43.6%). Only a few imagined feelings very sad or sad (6%) and very nervous or nervous (13.8%).

When presented with a scenario in which they had a toothache and went to the dentist, 28.7% of participants imagined feeling neither happy nor sad, and 31.7% imagined feeling very calm. A larger percentage than described before imagined feeling sad (19.8%) or very sad (10.9%) and nervous (16.8%) or very nervous (12.9%).

Participants imagined being happier, t (100) = 5.57, *p* < 0.001, *d* = 0.60, and calmer, t(100) = 4.05, *p* < 0.001, *d* = 0.37, when visiting the dentist without a toothache (respectively, *M* = 3.98; SD = 1.02 and *M* = 3.90; SD = 1.25) than when visiting the dentist with a toothache (respectively, *M* = 3.26; SD = 1.34 and *M* = 3.41; SD = 1.42).

A MANOVA was conducted entering how happy and calm participants imagined feeling when visiting the dentist (routine visit as well as when experiencing pain) as a criterion and sex and school year as predictors. Significant main effects of school year were found for feeling happy when visiting the dentist, in a routine visit (*F* (1, 97) = 9.36, *p* = 0.003, ƞp^2^ = 0.086), as well as when experiencing pain (*F* (1, 97) = 20.51, *p* < 0.001, ƞp^2^ = 0.175). Moreover, significant main effects of the school year were found for feeling calm when visiting the dentist, in a routine visit (*F* (1, 97) = 12.11, *p* = 0.001, ƞp^2^ = 0.111), as well as when experiencing pain (*F* (1, 97) = 15.62, *p* < 0.001, ƞp^2^ = 0.139). Children in the third grade, compared to children in the fourth grade, imagined feeling happier in a routine visit (respectively, *M* = 4.25; SD = 0.132 and *M* = 3.64; SD = 0.147) as well as when experiencing pain (*M* = 3.76; SD = 0.164 and *M* = 2.64; SD = 0.183). Similarly, children in the third grade imagined feeling calmer than fourth graders in a routine visit (respectively, *M* = 4.28; SD = 0.159 and *M* = 3.45; SD = 0.178) as well as when experiencing pain (respectively, *M* = 3.88; SD = 0.178 and *M* = 2.82; SD = 0.199). No main effect of sex was found (*F* < 0.83) as well as no significant interactions between sex and school year (*p* < 0.298).

Age was tested as a predictor of how happy and calm participants imagined feeling when visiting the dentist (routine visit as well as when experiencing pain). Age was a significant negative predictor of how calm (β = −0.233, t (99) = −2.38, *p* = 0.019, R^2^ = 0.054) participants imagined feeling in a routine visit but not how happy (β = −0.085, t (99) = −0.845, *p* = 0.400, R^2^ = 0.007). Conversely, age was a significant negative predictor of how happy (β = −0.207, t (99) = −2.11, *p* = 0.037, R^2^ = 0.043) participants imagined feeling at the dentist when experiencing pain but not how calm (β = −0.121, t (99) = −1.21, *p* = 0.228, R^2^ = 0.015).

To assess whether previous experiences had an effect on how participants felt when visiting the dentist (routine visit as well as when experiencing pain), a multivariate analysis of variance was run entering “Have you ever had a bad experience at the dentist?” as an independent variable and how happy and calm participants imagined feeling when visiting the dentist (routine visit as well as when experiencing pain) as dependent variables. No significant main effects were found (*p* > 0.276).

Knowledge concerning dental and oral hygiene

When asked what the right amount of toothpaste one should use when brushing their teeth was, most participants (70.3%) chose an option depicting an excess amount of toothpaste, and few participants (2.7%) chose the option corresponding to an amount that was too little. Only 27% of participants chose the option corresponding to the correct amount (see Table 3 for percentages of correct and incorrect responses for each question).

On average, participants considered that the maximum number of teeth an adult can have is 30 (*M* = 30.21; SD = 5.65). Responses ranged from 12 to 50 teeth. Many participants indicated the correct number of teeth (48.6%), and 51.3% indicated an incorrect number of teeth. Among those who gave an incorrect answer, 37.8% chose a lower number of teeth, and only 13.5% considered the maximum number of teeth to be more than 32.

When asked how many types of teeth a person could have, the average response was five (*M* = 4.80; SD = 4.22) and participants’ responses ranged from 2 to 30. Only 27% of participants chose the correct number (three). Of the 73% that wrote incorrect numbers, most participants (62.2%) chose a higher number of types of teeth, and 10.8% chose a lower number of types of teeth.

Participants considered that one should brush their teeth about three times per day (*M* = 3.14; SD = 0.58). Responses ranged from two to five times per day. Only 6.8% of participants chose the correct number of times (two). The remaining 93.2% of participants wrote that one should brush their teeth three or more times per day.

Regarding how long one should brush their teeth, on average, participants considered that it should last for about 5 min (*M* = 5.12; SD = 6.56). Responses ranged from 1 to 40 min of duration. From those participants that chose an incorrect duration, most were wrong by excess (56.8%), only 10.8% were wrong by default, and 32.4% of participants indicated the correct duration (2 min).

By counting the correct responses for the questions described above, a score of correct responses was obtained for each participant (Table 4). In total, 23% did not choose one correct response, and most participants only got one correct response (44.6%).

An ANOVA entering sex and school year as independent variables and the number of correct responses as a dependent variable was run. Only school year yielded a significant effect (*F* (1, 70) = 13.10, *p* = 0.001, ƞp^2^ = 0.158). Children in the third grade (*M* = 1.44; SD = 1.07) got more correct responses than those in the fourth grade (*M* = 0.79; SD = 1.15). No other effects were significant (*p* > 0.241).

Age was also found to be a negative predictor of the number of correct responses (β = −0.257, t (72) = −2.26, *p* = 0.027, R^2^ = 0.066).

To understand if knowledge about dental and oral health issues depended on previous experience, chi-square tests were performed. However, no significant effects were found (*p* > 0.329).

## 4. Discussion

The present project aims to provide conceptual and theoretical outcomes, giving evidence about children’s knowledge and attitudes about oral health and dental practices.

Children reported accurate and positive behaviors regarding dental and oral hygiene habits. Most of these children brush their teeth twice a day and every day, using toothpaste. However, only about half of them reported flossing or using mouthwash.

Regarding previous experiences with dental problems and oral treatments, a large majority had already gone to the dentist, but only some of them reported to have already had painful experiences, such as cavities, gum pain, bleeding or even a tooth taken out. A relevant aspect is that only 28% of these children reported having already had a negative and traumatic experience at the dentist.

In general, children’s opinions and beliefs about dentists were globally positive. Most of them have not only a positive attitude about them but they also generally trust in dentists and believe in the success of treatments/help if needed. Most children also reported positive emotions (i.e., feeling happy and calm) in an imaginary scenario of visiting the dentist. However, some of them also described feelings of fear, sadness and stress. The results suggested an effect of children’s age and educational level: older children reported more negative attitudes and emotions, as well as painful feelings about visiting the dentist.

Finally, and concerning children’s knowledge about dental and oral hygiene, the results were quite negative and worrying, ruled by ignorance and incorrect beliefs. Most children answered wrongly about the amount of toothpaste needed for a brushing once, its correct time/duration and daily frequency. In the same line, the majority of children also reported incorrect knowledge about quite basic questions, such as the maximum number of teeth an adult can have or even how many types of teeth are there. Regarding children’s knowledge responses, the results also showed the significant effect of educational level and age. Age was found to be a negative predictor of the number of correct responses. Thus, children in the third grade provided more correct responses than those in the fourth grade. A possible explanation for this significant effect could be related to the current educational program in the second grade of Portuguese schools, where younger children learn about several human body parts, in particular the oral cavity.

Future studies should be conducted using a larger sample of children from different contextual statuses. These could be the main limitations of the present study: the sample size and the similar economic status (all of our children are included in a medium economic familiar cultural level). A larger sample might allow conducting a more complex statistical analysis and test more significant effects. Another suggestion is to include other social and contextual variables, such as the economic status (SES) of the participants. The literature suggests the disadvantaged SES contexts as risk indicators for dental and oral problems. In fact, several studies showed that bad oral health has a significant relation with children’s SES [16,17,18]. Early dental problems are usually described as associated with sociodemographic factors, dietary and oral health habits [19].

Dental stigma, fear and phobia comprise a complex and multifactorial current problem [20]. The research evidence suggests that it might be related to both endogenous (e.g., personality traits) and exogenous (e.g., information through media; previous traumatic experiences; modeling and vicarious learning through significant others) factors [20,21,22]. The evidence also suggests that this negative approach have clear and practical implications, and it should be conceptualized as consisting of a triad: attitudes (i.e., prejudice, fear), knowledge (i.e., ignorance and lack of valid and appropriate knowledge) and behavior (i.e., avoidance, refuse) [20,21].

While there is increasing knowledge about health literacy in adult populations, less evidence and interventions are available and implemented with children [21,22]. According to the literature, children with access to clear information about health entertain fewer negative worries, more knowledge and are able to address these issues in a more realistic and appropriate manner. This evidence could be explained by the Information Provision Model (IPM) that was designed to integrate the various processes involved in information provision, such as the self-regulation theory and schema/script theories, as well as the role that individual difference aspects (e.g., age, temperament, coping styles) may play in how children respond to such information. Briefly, this model theorizes that by providing information about health procedures, children may be able to identify the most appropriate schemata to cope with that event [23]. Accurate information may contribute to the decrease in fear and anxiety related to anticipation of medical-health events and lead to positive outcomes, minimized uncertainties, reduced threat and fear perception and inconsistencies between unreal expectation and the reality of dental procedures. On the contrary, children without this accurate information may activate inappropriate schemas related to healthcare settings, increasing their negative responses (e.g., worries, fears, and anxiety), resulting in fewer realistic expectations and future avoidance behaviors [23].

Since the last century, children’s access to information about body and health procedures has been widely advocated as one of their most fundamental rights [24]. Information provision is a crucial aspect of preparing children for pediatric health procedures [23,25].

Nowadays, the use of serious ludic materials (e.g., games, activities, books) specifically designed for health and medical procedures education is growing rapidly [26,27,28]. Play with ludic games requires a deep and interactive involvement of players, and it is an effective tool to provide learning experiences and improve health-related behaviors through an entertaining and motivating format [28,29,30]. Previous studies have also shown that interventions to provide educational information are especially effective in children at the concrete operational stage, which, according to Piaget’s theory (1963), includes children aged between 7/8 and 12 years [24,29,30,31].

## 5. Conclusions

Given the conceptual and theoretical outcomes regarding children’s knowledge and attitudes about oral health and dental practices, the present project will contribute to the development of educational and innovative materials about oral health and hygiene practices—“An Adventure about Oral Health” (edited by Ideias com Historia). We will bear practical outcomes by developing educational and ludic materials (in both board-game and activities’ book formats) about these oral and dental issues.

In summary, children’s responses collected in the present study will be addressed in the educational programs “An Adventure about Oral Health,” intending to increase positive attitudes and behaviors related to oral health, as well as promote greater knowledge about this important subject.

## 6. Patents

This section is not mandatory but may be added if there are patents resulting from the work reported in this manuscript.

## Figures and Tables

**Table 1 healthcare-09-00416-t001:** Frequencies and percentages of children’s responses about their dental habits and behaviors.

	Frequency	Percent (%)
“Do you brush your teeth every day?”		
Yes	100	99.0
No	1	1.0
“How many times per day do you brush your teeth?”		
1 time	8	7.9
2 times	48	47.5
3 or more times	43	42.6
Missing	2	2.0
“Do you have an adult helping you brush your teeth?”		
Yes	2	2.0
Sometimes	9	8.9
No	101	89.1
“Do you use toothpaste when brushing your teeth?”		
Yes	98	97.0
Sometimes	1	1.0
No	1	1.0
Missing	1	1.0
“Do you floss your teeth?”		
Yes	15	14.9
Sometimes	37	36.6
No	48	47.5
Missing	1	1.0
“Do you use mouthwash?”		
Yes	27	26.7
Sometimes	32	31.7
No	42	41.6
“Have you ever been to the dentist?”		
Yes	88	87.1
No	11	10.9
Missing	2	2.0
“Have you ever had a cavity?”		
Yes	46	45.5
No	55	54.5
“Have you ever had gum pain?”		
Yes	54	53.3
No	46	45.5
Missing	1	1.0
“Did you ever bleed from your gums?”		
Yes	76	75.2
No	25	24.8
“Have you ever had a tooth taken out?”		
Yes	69	68.3
No	32	31.7
“Have you ever had a bad experience at the dentist?”		
Yes	27	26.7
No	71	70.3
Missing	3	3.0

**Table 2 healthcare-09-00416-t002:** Frequencies and percentages of children’s responses about their attitudes and opinions about dentists.

	Frequency	Percent (%)
“What is your opinion of dentists?”		
Do not like them	11	10.9
Do not know	17	16.8
Like them	73	72.3
“Imagine you have a problem in your teeth, and you go to the dentist. After the dentist appointment, do you think you will get…”		
Better	93	92.1
Same	6	5.9
Worse	2	2.0

**Table 3 healthcare-09-00416-t003:** Percentage of correct responses by question.

	Incorrect	Correct	Incorrect
	by Default		by Excess
Amount of toothpaste	2.7	27	70.3
Maximum number of teeth (32)	37.8	48.6	13.5
Types of teeth (3)	10.8	27	62.2
Teeth brushing—how many times a day (2)	0	6.8	93.2
Teeth brushing—for how long (2 mn)	10.8	32.4	56.8

**Table 4 healthcare-09-00416-t004:** Percentage of total correct responses.

Number of Correct Responses	%
0	23.0
1	44.6
2	27.0
3	5.4

## Data Availability

The data presented in this study are available on request from the corresponding author. The data are not publicly available due to privacy reasons.

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
