# Peer review of "Children’s Attitudes and Behaviors about Oral Health and Dental Practices"

_healthcare, 2021, doi:10.3390/healthcare9040416_

Round 1
Reviewer 1 Report
the study is interesting and rise up a very important problem.
However there are some correction to be performed:
Abstract should be rewritten being more in line with the article.
Both abstract and conclusions mention a program inappropriately, so You have two choices: explain the program in the intro and the relationship with the study or take it out.
Statistical analysis paragraph is missing in the methods:
Which program You used, the test You performed (which you described in results, but it should be placed in the methods):
Author Response
Dear Reviewer #1:
Thank you very much for all your comments and suggestions. It was a pleasure to address all the comments provided by you, and we think that the manuscript has greatly benefited from this revision. Overall, we appreciate the time and efforts expended in reviewing our manuscript.
We respond below: we first included your comments repeated in italics, and after each them we explain the way we addressed it.
- Abstract should be rewritten being more in line with the article.
Response: Done. We reviewed the abstract, describing the main goals of the study in a more precise way
- Both abstract and conclusions mention a program inappropriately, so You have two choices: explain the program in the intro and the relationship with the study or take it out.
Response: Done. We decided to keep the reference to the program “An adventure about oral health” because, in fact, the present project was the scientific support for its development. We include more information about it and we also explain this relevance.
- Statistical analysis paragraph is missing in the methods: Which program You used, the test You performed (which you described in results, but it should be placed in the methods).
Response: Done. We include a new sentence about the statistical program used, as well as we replace the analysis conducted on the methods section.

Reviewer 2 Report
Some clarification of the manuscript is necessary:
1) In the abstract it is necessary to describe the aims of the study more precisely.
2) In the introduction, the literature references are very limited.
3) Why has the sample not been expanded in children during 2020?
4) How was the sample size determined?
5) Why was an age range between 8 and 10 years indicated?
6) Has the questionnaire used in this study been previously validated?
7) Why has the reference of the Bioethics Committee not been indicated in the manuscript?
8) Why have the limitations of the study not been described?
9) The bibliographical references are not described according to the journal's regulations.
10) Why was the STROBE list not included in the manuscript?
Author Response
Dear Reviewer #2:
Thank you very much for all your comments and suggestions. It was a pleasure to address all the comments provided by you, and we think that the manuscript has greatly benefited from this revision. Overall, we appreciate the time and efforts expended in reviewing our manuscript.
We respond below: we first included your comments repeated in italics, and after each them we explain the way we addressed it.
1) In the abstract it is necessary to describe the aims of the study more precisely.
Response: Done. We reviewed the abstract, describing the aims of the study in a more precise way
2) In the introduction, the literature references are very limited.
Response: We reformulated the introduction part and added some references in the discussion part. Please let us know if you would like more information.
3) Why has the sample not been expanded in children during 2020?
Response: Due to restrictions imposed by the pandemic COVID19 situation in Portugal, in particular with the closure of the schools, it was not possible to collect more data. We decided to keep the current sample (although relatively small size) but we guarantee that its collection was made under the same conditions.
4) How was the sample size determined?
Response: We used the G*Power 3.2.3 software analysis. We included a paragraph in the Method’s section about the sample size.
5) Why was an age range between 8 and 10 years indicated?
Response: The age range between 8-10 years was chosen to ensure that all the sample would be at the same concrete operational stage of development, according to Piaget’s theory (1963), and so, children will have similar interests and experiences.
On the other hand, and according to the literature, the educational programs developed based on this study, “An adventure about oral health” are especially effective in children at the concrete operational stage, which according to Piaget’s theory (1963) includes children aged between 7/8 to 12 years [24, 25]
6) Has the questionnaire used in this study been previously validated?
Response: Done. We include a new paragraph in the Methods’ section, explaining that the questionnaires used to obtain information about children’s dental and oral issues were constructed using the same questions included on both questionnaires Modified Child Dental Anxiety Scale (MCDASF) and Children's Fear Survey Schedule – Dental Subscale (CFSS-DS), which were both largely tested and validated in 5–12 year-old children across the world.
7) Why has the reference of the Bioethics Committee not been indicated in the manuscript?
Response: The present study was validated and authorized by several Ethical Committees (ie., University, schools where the study was conducted), as well as by the Portuguese Ministry of Education. We included more information about it on Materials and Methods section.
8) Why have the limitations of the study not been described?
Response: We have now included, on Discussion section, the two main limitations of the present study: the sample size and its similar economic status. We also discussed about these limitations, justifying them and giving some suggestions.
9) The bibliographical references are not described according to the journal's regulations.
Response: Done. We correct the information and now the references are described according to the journal’s rules.
10) Why was the STROBE list not included in the manuscript?
Response: Taking into account the relevance of the STROBE list, we try to organize our manuscript according to this checklist’s suggestions. Please let us know if there is some point that you would like to see clearer.

Round 2
Reviewer 2 Report
Authors must remove the DOI link in the bibliographical references.
The title of bibliographical references should be in italics.
Authors should pay attention to these details when writing a scientific paper.